# ON CONVERGENCE AND STABILITY OF GANS

## ABSTRACT

We propose studying GAN training dynamics as regret minimization, which is in contrast to the popular view that there is consistent minimization of a divergence between real and generated distributions. We analyze the convergence of GAN training from this new point of view to understand why mode collapse happens. We hypothesize the existence of undesirable *local equilibria* in this non-convex game to be responsible for mode collapse. We observe that these local equilibria often exhibit sharp gradients of the discriminator function around some real data points. We demonstrate that these degenerate local equilibria can be avoided with a gradient penalty scheme called DRAGAN. We show that DRAGAN enables faster training, achieves improved stability with fewer mode collapses, and leads to generator networks with better modeling performance across a variety of architectures and objective functions.

## 1 INTRODUCTION

Generative modeling involves taking a set of samples drawn from an unknown data generating distribution $P_{real}$ and finding an estimate $P_{model}$ that closely resembles it. Generative adversarial networks (GAN) (Goodfellow et al., 2014) is a powerful framework used for fitting *implicit* generative models. The basic setup consists of two networks, the generator and the discriminator, playing against each other in a repeated zero-sum game setting. The goal here is to reach an equilibrium where $P_{real}$, $P_{model}$ are close, and the alternating gradient updates procedure (AGD) is used to achieve this. However, this process is highly unstable and often results in mode collapse (Goodfellow, 2017). This calls for an deeper investigation into training dynamics of GANs.

In this paper, we propose studying GAN training dynamics as a repeated game in which both the players are using no-regret algorithms (Cesa-Bianchi & Lugosi, 2006) and discuss how AGD [1] falls under this paradigm. In contrast, much of the theory (Goodfellow et al., 2014; Arjovsky & Bottou, 2017) and recent developments (Nowozin et al., 2016; Arjovsky et al., 2017; Gulrajani et al., 2017) are based on the unrealistic assumption that the discriminator is playing optimally (in the function space) at each step and as a result, there is consistent minimization of a divergence between real and generated distributions. This corresponds to at least one player using the *best-response* algorithm (in the function space), and the resulting game dynamics can be completely different in both these cases (Nisan et al., 2007). Thus, there is a clear disconnect between theoretical arguments used as motivation in recent literature and what actually happens in practice.

We would like to point out that the latter view can still be useful for reasoning about the asymptotic equilibrium situation but we argue that regret minimization is the more appropriate way to think about GAN training dynamics. So, we analyze the convergence of GAN training from this new point of view to understand why mode collapse happens. We start with a short analysis of the artificial convex-concave case of the GAN game in section 2.2. This setting has a unique solution and guaranteed convergence (of averaged iterates) using no-regret algorithms can be shown with standard arguments from game theory literature. Here, we make explicit, the critical (previously not widely known) connection between AGD used in GAN training and regret minimization. This immediately yields a novel proof for the asymptotic convergence of GAN training, in the non-parametric limit. Prior to our work, such a result (Goodfellow et al., 2014) required a strong assumption that the discriminator is optimal at each step.

---

[1]Most of our analysis applies to the simultaneous gradient updates procedure as well

However, these convergence results do not hold when the game objective function is non-convex, which is the practical case when deep neural networks are used. In non-convex games, global regret minimization and equilibrium computation are computationally hard in general. Recent game-theoretic literature indicates that AGD can end up cycling (Mertikopoulos et al., 2017) or converging to a (potentially bad) local equilibrium, under some conditions (Hazan et al., 2017). We hypothesize these to be the reasons for cycling and mode collapse observed during GAN training, respectively (section 2.3). In this work, we do not explore the cycling issue but focus our attention on the mode collapse problem. In contrast to our hypothesis, the prevalent view of mode collapse and instability (Arjovsky & Bottou, 2017) is that it results from attempting to minimize a *strong* divergence during training. However, as we argued earlier, GAN training with AGD does not consistently minimize a divergence and therefore, such a theory is not suitable to discuss convergence or to address the stability issue.

Next, if mode collapse is indeed the result of an undesirable local equilibrium, a natural question then is how we can avoid it? We make a simple observation that, in the GAN game, mode collapse situations are often accompanied by sharp gradients of the discriminator function around some real data points (section 2.4). Therefore, a simple strategy to mitigate mode collapse is to regularize the discriminator so as to constrain its gradients in the ambient data space. We demonstrate that this improves the stability using a toy experiment with one hidden layer neural networks. This gives rise to a new explanation for why WGAN and gradient penalties might be improving the stability of GAN training – they are mitigating the mode collapse problem by keeping the gradients of the discriminator function small in data space. From this motivation, we propose a training algorithm involving a novel gradient penalty scheme called DRAGAN (Deep Regret Analytic Generative Adversarial Networks) which enables faster training, achieves improved stability and modeling performance (over WGAN-GP (Gulrajani et al., 2017) which is the state-of-the-art stable training procedure) across a variety of architectures and objective functions.

Below, we provide a short literature review. Several recent works focus on stabilizing the training of GANs. While some solutions (Radford et al., 2015; Salimans et al., 2016) require the usage of specific architectures (or) modeling objectives, some (Che et al., 2016; Zhao et al., 2016) significantly deviate from the original GAN framework. Other promising works in this direction (Metz et al., 2016; Arjovsky et al., 2017; Qi, 2017; Gulrajani et al., 2017) impose a significant computational overhead. Thus, a fast and versatile method for consistent stable training of GANs is still missing in the literature. Our work is aimed at addressing this.

To summarize, our contributions are as follows:

- We propose a new way of reasoning about the GAN training dynamics - by viewing AGD as regret minimization.

- We provide a novel proof for the asymptotic convergence of GAN training in the non-parametric limit and it does not require the discriminator to be optimal at each step.

- We discuss how AGD can converge to a potentially bad local equilibrium in non-convex games and hypothesize this to be responsible for mode collapse during GAN training.

- We characterize mode collapse situations with sharp gradients of the discriminator function around some real data points.

- A novel gradient penalty scheme called DRAGAN is introduced based on this observation and we demonstrate that it mitigates the mode collapse issue.

## 2 THEORETICAL ANALYSIS OF GAN TRAINING DYNAMICS

We start with a brief description of the GAN framework (section 2.1). We discuss guaranteed convergence in the artificial convex-concave case using no-regret algorithms, and make a critical connection between GAN training process (AGD) and regret minimization (section 2.2). This immediately yields a novel proof for the asymptotic convergence of GAN training in the non-parametric limit. Then, we consider the practical non-convex case and discuss how AGD can converge to a potentially bad local equilibrium here (section 2.3). We characterize mode collapse situations with sharp gradients of the discriminator function around real samples and this provides an effective strategy to avoid them. This naturally leads to the introduction of our gradient penalty

scheme DRAGAN (section 2.4). We end with a discussion and comparison with other gradient penalties in the literature (section 2.5).

## 2.1 BACKGROUND

The GAN framework can be viewed as a repeated zero-sum game, consisting of two players - the *generator*, which produces synthetic data given some noise source and the *discriminator*, which is trained to distinguish generator's samples from the real data. The generator model G is parameterized by $\phi$, takes a noise vector $\mathbf{z}$ as input, and produces a synthetic sample $G_\phi(\mathbf{z})$. The discriminator model D is parameterized by $\theta$, takes a sample $\mathbf{x}$ as input and computes $D_\theta(\mathbf{x})$, which can be interpreted as the probability that $\mathbf{x}$ is real.

The models G, D can be selected from any arbitrary class of functions – in practice, GANs typical rely on deep networks for both. Their cost functions are defined as

$$J^{(D)}(\phi, \theta) := -\mathbb{E}_{x \sim p_{real}} \log D_\theta(x) - \mathbb{E}_{\mathbf{z}} \log(1 - D_\theta(G_\phi(z))), \text{ and}$$

$$J^{(G)}(\phi, \theta) := -J^{(D)}(\phi, \theta)$$

And the complete game can be specified as -

$$\min_\phi \max_\theta \left\{ J(\phi, \theta) = \mathbb{E}_{x \sim p_{real}} \log D_\theta(x) + \mathbb{E}_{\mathbf{z}} \log(1 - D_\theta(G_\phi(z))) \right\}$$

The generator distribution $P_{model}$ asymptotically converges to the real distribution $P_{real}$ if updates are made in the function space and the discriminator is optimal at each step (Goodfellow et al., 2014).

## 2.2 CONVEX-CONCAVE CASE AND NO-REGRET ALGORITHMS

According to Sion's theorem (Sion, 1958), if $\Phi \subset \mathbb{R}^m$, $\Theta \subset \mathbb{R}^n$ such that they are compact and convex sets, and the function $J : \Phi \times \Theta \to \mathbb{R}$ is convex in its first argument and concave in its second, then we have -

$$\min_{\phi \in \Phi} \max_{\theta \in \Theta} J(\phi, \theta) = \max_{\theta \in \Theta} \min_{\phi \in \Phi} J(\phi, \theta)$$

That is, an equilibrium is guaranteed to exist in this setting where players' payoffs correspond to the unique value of the game (Neumann, 1928).

A natural question then is how we can find such an equilibrium. A simple procedure that players can use is best-response algorithms (BRD). In each round, best-responding players play their optimal strategy given their opponent's current strategy. Despite its simplicity, BRD are often computationally intractable and they don't lead to convergence even in simple games. In contrast, a technique that is both efficient and provably works is regret minimization. If both players update their parameters using no-regret algorithms, then it is easy to show that their averaged iterates will converge to an equilibrium pair (Nisan et al., 2007). Let us first define no-regret algorithms.

**Definition 2.1** (**No-regret algorithm**). Given a sequence of convex loss functions $L_1, L_2, \dots :$ $K \to \mathbb{R}$, an algorithm that selects a sequence of $k_t$'s, each of which may only depend on previously observed $L_1, \dots, L_{t-1}$, is said to have *no regret* if $\frac{R(T)}{T} = o(1)$, where we define

$$R(T) := \sum_{t=1}^T L_t(k_t) - \min_{k \in K} \sum_{t=1}^T L_t(k)$$

We can apply no-regret learning to our problem of equilibrium finding in the GAN game $J(\cdot, \cdot)$ as follows. The generator imagines the function $J(\cdot, \theta_t)$ as its loss function on round $t$, and similarly the discriminator imagines $-J(\phi_t, \cdot)$ as its loss function at $t$. After $T$ rounds of play, each player computes the average iterates $\bar{\phi}_T := \frac{1}{T} \sum_{t=1}^T \phi_t$ and $\bar{\theta}_T := \frac{1}{T} \sum_{t=1}^T \theta_t$. If $V^*$ is the equilibrium value of the game, and the players suffer regret $R_1(T)$ and $R_2(T)$ respectively, then one can show using standard arguments (Freund & Schapire, 1999) that -

$$V^* - \frac{R_2(T)}{T} \le \max_{\theta \in \Theta} J(\bar{\phi}_T, \theta) - \frac{R_2(T)}{T} \le \min_{\phi \in \Phi} J(\phi, \bar{\theta}_T) + \frac{R_1(T)}{T} \le V^* + \frac{R_1(T)}{T}.$$

In other words, $\bar{\theta}_T$ and $\bar{\phi}_T$ are "almost optimal" solutions to the game, where the "almost" approximation factor is given by the average regret terms $\frac{R_1(T) + R_2(T)}{T}$. Under the no-regret

condition, the former will vanish, and hence we can guarantee convergence in the limit. Next, we define a popular family of no-regret algorithms.

**Definition 2.2 (Follow The Regularized Leader).** FTRL (Hazan et al., 2016) selects $k_t$ on round $t$ by solving for $\arg\min_{k \in K}\{\sum_{s=1}^{t-1} L_s(k) + \frac{1}{\eta}\Omega(k)\}$, where $\Omega(\cdot)$ is some convex regularization function and $\eta$ is a learning rate.

**Remark:** Roughly speaking, if you select the regularization as $\Omega(\cdot) = \frac{1}{2}\|\cdot\|^2$, then FTRL becomes the well-known online gradient descent or OGD (Zinkevich, 2003). Ignoring the case of constraint violations, OGD can be written in a simple iterative form: $k_t = k_{t-1} - \eta\nabla L_{t-1}(k_{t-1})$.

The typical GAN training procedure using alternating gradient updates (or simultaneous gradient updates) is almost this - both the players applying online gradient descent. Notice that the min/max objective function in GANs involves a stochastic component, with two randomized inputs given on each round, $x$ and $z$ which are sampled from the data distribution and a standard multivariate normal, respectively. Let us write $J_{x,z}(\phi, \theta) := \log D_\theta(x) + \log(1 - D_\theta(G_\phi(z)))$. Taking expectations with respect to $\mathbf{x}$ and $\mathbf{z}$, we define the full (non-stochastic) game as $J(\phi, \theta) = \mathbb{E}_{\mathbf{x},\mathbf{z}}[J_{x,z}(\phi, \theta)]$. But the above online training procedure is still valid with stochastic inputs. That is, the equilibrium computation would proceed similarly, where on each round we sample $x_t$ and $z_t$, and follow the updates

$$\phi_{t+1} \leftarrow \phi_t - \eta\nabla_\phi J_{x_t,z_t}(\phi_t, \theta_t). \quad \text{and} \quad \theta_{t+1} \leftarrow \theta_t + \eta'\nabla_\theta J_{x_t,z_t}(\phi_t, \theta_t)$$

On a side note, a benefit of this stochastic perspective is that we can get a generalization bound on the mean parameters $\bar{\phi}_T$ after $T$ rounds of optimization. The celebrated "online-to-batch conversion" (Cesa-Bianchi et al., 2004) implies that $\mathbb{E}_{\mathbf{x},\mathbf{z}}[J_{x,z}(\bar{\phi}_T, \theta)]$, for any $\theta$, is no more than the optimal value $\mathbb{E}_{\mathbf{x},\mathbf{z}}[J_{x,z}(\phi^*, \theta)]$ plus an "estimation error" bounded by $\mathbb{E}\left[\frac{R_1(T)+R_2(T)}{T}\right]$, where the expectation is taken with respect to the sequence of samples observed along the way, and any randomness in the algorithm. Analogously, this applies to $\bar{\theta}_T$ as well. A limitation of this result, however, is that it requires a fresh sample $x_t$ to be used on every round.

To summarize, we discussed in this subsection about how the artificial convex-concave case is easy to solve through regret minimization. While this is a standard result in game theory and online learning literature, it is not widely known in the GAN literature. For instance, Salimans et al. (2016) and Goodfellow (2017) discuss a toy game which is convex-concave and show cycling behavior. But, the simple solution in that case is to just average the iterates. Further, we made explicit, the critical connection between regret minimization and alternating gradient updates procedure used for GAN training. Now, Goodfellow et al. (2014) argue that, if $G$ and $D$ have enough capacity (in the non-parametric limit) and updates are made in the function space, then the GAN game can be considered convex-concave. Thus, our analysis based on regret minimization immediately yields a novel proof for the asymptotic convergence of GANs, without requiring that the discriminator be optimal at each step.

Moreover, the connection between regret minimization and GAN training process gives a novel way to reason about its dynamics. In contrast, the popular view of GAN training as consistently minimizing a divergence arises if the discriminator uses BRD (in the function space) and thus, it has little to do with the actual training process of GANs. As a result, this calls into question the motivation behind many recent developments like WGAN and gradient penalties among others, which improve the training stability of GANs. In the next subsection, we discuss the practical non-convex case and why training instability arises. This provides the necessary ideas to investigate mode collapse from our new perspective.

### 2.3 NON-CONVEX CASE AND LOCAL EQUILIBRIA

In practice, we choose $G$, $D$ to be deep neural networks and the function $J(\phi, \theta)$ need not be convex-concave anymore. The nice properties we had in the convex-concave case like the existence of a unique solution and guaranteed convergence through regret minimization no longer hold. In fact, regret minimization and equilibrium computation are computationally hard in general non-convex settings. However, analogous to the case of non-convex optimization (also intractable) where we focus on finding local minima, we can look for tractable solution concepts in non-convex games.

Recent work by Hazan et al. (2017) introduces the notion of *local regret* and shows that if both the players use a *smoothed* variant of OGD to minimize this quantity, then the non-convex game converges to some form of local equilibrium, under mild assumptions. The usual training procedure of GANs (AGD) corresponds to using a *window size* of 1 in their formulation. Thus, GAN training will eventually converge (approximately) to a local equilibrium which is described below or the updates will cycle. We leave it to future works to explore the equally important cycling issue and focus here on the former case.

**Definition 2.3** (**Local Equilibrium**). A pair $(\phi^*, \theta^*)$ is called an $\epsilon$-approximate local equilibrium if it holds that

$$\forall \phi^{'}, \|\phi^{'} - \phi^*\| \leq \eta : J(\phi^*, \theta^*) \leq J(\phi^{'}, \theta^*) + \epsilon$$
$$\forall \theta^{'}, \|\theta^{'} - \theta^*\| \leq \eta : J(\phi^*, \theta^*) \geq J(\phi^*, \theta^{'}) - \epsilon$$

That is, in a local equilibrium, both the players do not have much of an incentive to switch to any other strategy within a small neighborhood of their current strategies. Now, we turn our attention to the mode collapse issue which poses a significant challenge to the GAN training process. The training is said to have resulted in mode collapse if the generator ends up mapping multiple **z** vectors to the same output **x**, which is assigned a high probability of being real by the discriminator (Goodfellow, 2017). We hypothesize this to be the result of the game converging to bad local equilibria.

The prevalent view of mode collapse and instability in GAN training (Arjovsky & Bottou, 2017) is that it is caused due to the supports of real and model distributions being disjoint or lying on low-dimensional manifolds. The argument is that this would result in strong distance measures like KL-divergence or JS-divergence getting maxed out, and the generator cannot get useful gradients to learn. In fact, this is the motivation for the introduction of WGAN (Arjovsky et al., 2017). But, as we argued earlier, GAN training does not consistently minimize a divergence as that would require using intractable best-response algorithms. Hence, such a theory is not suitable to discuss convergence or to address the instability of GAN training. Our new view of GAN training process as regret minimization is closer to what is used in practice and provides an alternate explanation for mode collapse - the existence of undesirable local equilibria. The natural question now is how we can avoid them?

## 2.4 Mode collapse and Gradient Penalties

The problem of dealing with multiple equilibria in games and how to avoid undesirable ones is an important question in algorithmic game theory (Nisan et al., 2007). In this work, we constrain ourselves to the GAN game and aim to characterize the undesirable local equilibria (mode collapse) in an effort to avoid them. In this direction, after empirically studying multiple mode collapse cases, we found that it is often accompanied by the discriminator function having sharp gradients around some real data points (See Figure 1 [2]). This intuitively makes sense from the definition of mode collapse discussed earlier. Such sharp gradients encourage the generator to map multiple $z$ vectors to a single output $x$ and lead the game towards a degenerate equilibrium. Now, a simple strategy to mitigate this failure case would be to regularize the discriminator using the following penalty -

$$\lambda \cdot \mathbb{E}_{x \sim P_{real}, \delta \sim N_d(0, cI)} \left[ \|\nabla_{\mathbf{x}} D_\theta(x + \delta)\|^2 \right]$$

This strategy indeed improves the stability of GAN training. We show the results of a toy experiment with one hidden layer neural networks in Figure 2 and Figure 3 to demonstrate this. This partly explains the success of WGAN and gradient penalties in the recent literature (Gulrajani et al., 2017; Qi, 2017), and why they improve the training stability of GANs, despite being motivated by reasoning based on unrealistic assumptions. However, we noticed that this scheme in its current form can be brittle and if over-penalized, the discriminator can end up assigning both a real point $x$ and noise $x + \delta$, the same probability of being real. Thus, a better choice of penalty is -

$$\lambda \cdot \mathbb{E}_{x \sim P_{real}, \delta \sim N_d(0, cI)} \left[ \max \left( 0, \|\nabla_{\mathbf{x}} D_\theta(x + \delta)\|^2 - k \right) \right]$$

Finally, due to practical optimization considerations (this has also been observed in Gulrajani et al. (2017)), we instead use the penalty shown below in all our experiments.

$$\lambda \cdot \mathbb{E}_{x \sim P_{real}, \delta \sim N_d(0, cI)} \left[ \|\nabla_{\mathbf{x}} D_\theta(x + \delta)\| - k \right]^2 \tag{1}$$

---

[2]At times, stochasticity seems to help in getting out of the *basin of attraction* of a bad equilibrium

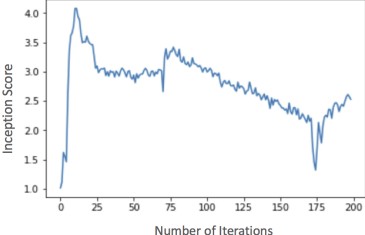 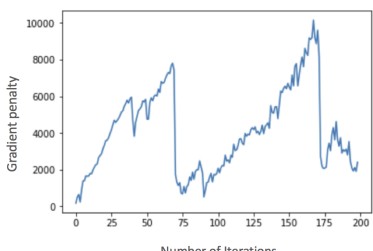

Figure 1: One hidden layer networks as $G$ and $D$ (MNIST). On the left, we plot inception score against time for vanilla GAN training and on the right, we plot the squared norm of discriminator's gradients around real data points for the same experiment. Notice how this quantity changes before, during and after mode collapse events.

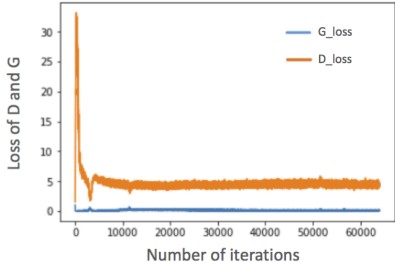 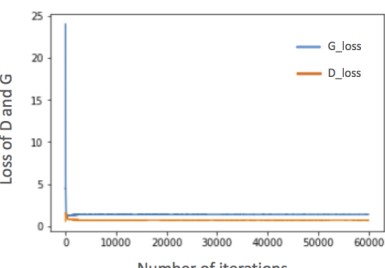

Figure 2: One hidden layer networks as $G$ and $D$ (MNIST). On the left, losses for both the players are shown for vanilla GAN training and on the right, we added a regularization term to penalize the gradients of $D(x)$ around real data points. Notice the improved stability.

This still works as long as small perturbations of real data, $x + \delta$ are likely to lie off the data-manifold, which is true in the case of image domain and some other settings. Because, in these cases, we do want our discriminator to assign different probabilities of being real to training data and noisy samples. We caution the practitioners to keep this important point in mind while making their choice of penalty. All of the above schemes have the same effect of constraining the norm of discriminator's gradients around real points to be small and can therefore, mitigate the mode collapse situation. We refer to GAN training using these penalty schemes or heuristics as the DRAGAN algorithm.

Additional details:

- We use the vanilla GAN objective in our experiments, but our penalty improves stability using other objective functions as well. This is demonstrated in section 3.3.
- The penalty scheme used in our experiments is the one shown in equation 1.
- We use small pixel-level noise but it is possible to find better ways of imposing this penalty. However, this exploration is beyond the scope of our paper.
- The optimal configuration of the hyperparameters for DRAGAN depends on the architecture, dataset and data domain. We set them to be $\lambda \sim 10$, $k = 1$ and $c \sim 10$ in most of our experiments.

## 2.5 COUPLED VS LOCAL PENALTIES

Several recent works have also proposed regularization schemes which constrain the discriminator's gradients in the ambient data space, so as to improve the stability of GAN training. Despite being from different motivations, WGAN-GP and LS-GAN are closely related approaches to ours. First, we show that these two approaches are very similar, which is not widely known in the literature. Qi (2017) introduced LS-GAN with the idea of maintaining a margin between losses assigned to real and

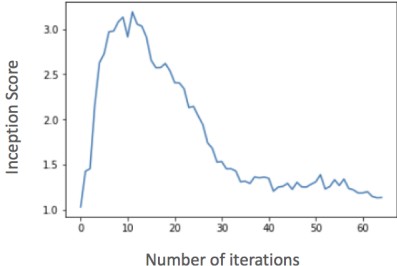 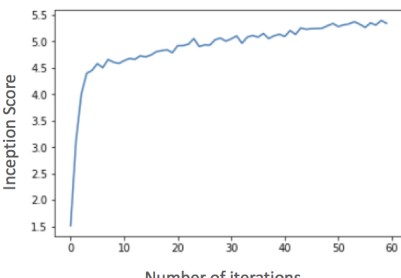

Figure 3: One hidden layer networks as $G$ and $D$ (MNIST). On the left, inception score plot is shown for vanilla GAN training and on the right, we added a regularization term to penalize the gradients of $D(x)$ around real data points. Notice how mode collapse is mitigated.

fake samples. Further, they also impose Lipschitz constraint on $D$ and the two conditions together result in a situation where the following holds for any real and fake sample pair (roughly) -

$$D_\theta(x) - D_\theta(G_\phi(z)) \approx ||x, G_\phi(z)|| \qquad (2)$$

The authors argue that the resulting discriminator function would have non-vanishing gradients almost everywhere between real and fake samples (section 6 of Qi (2017)). Next, Gulrajani et al. (2017) proposed an extension to address various shortcomings of the original WGAN and they impose the following condition on $D$ -

$$||\nabla_x D_\theta(\hat{x})|| \approx 1 \qquad (3)$$

where $\hat{x} = (\epsilon)x + (1-\epsilon)G_\phi(z)$ is some point on the line between a real and a fake sample, both chosen independently at random. This leads to $D$ having norm-1 gradients almost everywhere between real and fake samples. Notice that this behavior is very similar to that of LS-GAN's discriminator function. Thus, WGAN-GP is a slight variation of the original LS-GAN algorithm and we refer to these methods as "coupled penalties".

On a side note, we also want to point out that WGAN-GP's penalty doesn't actually follow from KR-duality as claimed in their paper. By Lemma 1 of Gulrajani et al. (2017), the optimal discriminator $D^*$ will have norm-1 gradients (almost everywhere) only between those $x$ and $G_\phi(z)$ pairs which are sampled from the optimal coupling or joint distribution $\pi^*$. Therefore, there is no basis for WGAN-GP's penalty (equation 3) where arbitrary pairs of real and fake samples are used. This fact adds more credence to our theory regarding why gradient penalties might be mitigating mode collapse.

The most important distinction between coupled penalties and our methods is that we only impose gradient constraints in local regions around real samples. We refer to these penalty schemes as "local penalties". Coupled penalties impose gradient constraints between real and generated samples and we point out some potential issues that arise from this:

- With adversarial training finding applications beyond fitting implicit generative models, penalties which depend on generated samples can be prohibitive.

- The resulting class of functions when coupled penalties are used will be highly restricted compared to our method and this affects modeling performance. We refer the reader to Figure 4 and appendix section 5.2.2 to see this effect.

- Our algorithm works with AGD, while WGAN-GP needs multiple inner iterations to optimize D. This is because the generated samples can be anywhere in the data space and they change from one iteration to the next. In contrast, we consistently regularize $D_\theta(x)$ only along the real data manifold.

To conclude, appropriate constraining of the discriminator's gradients can mitigate mode collapse but we should be careful so that it doesn't have any negative effects. We pointed out some issues with coupled penalties and how local penalties can help. We refer the reader to section 3 for further experimental results.

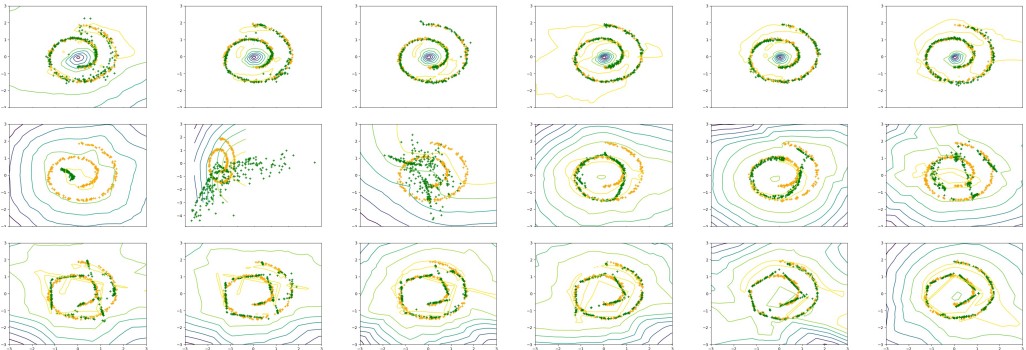

Figure 4: Swissroll experiment (different phases of training) - Vanilla GAN (top), WGAN-GP (middle), and DRAGAN (bottom). Real samples are marked orange and generated samples are green. Level sets of $D_\theta(x)$ are shown in the background where yellow is high and purple is low.

## 3 EXPERIMENTAL RESULTS

In section 3.1, we compare the modeling performance of our algorithm against vanilla GAN and WGAN variants in the standard DCGAN/CIFAR-10 setup. Section 3.2 demonstrates DRAGAN's improved stability across a variety of architectures. In section 3.3, we show that our method also works with other objective functions. Appendix contains samples for inspection, some of the missing plots and additional results. Throughout, we use inception score (Salimans et al., 2016) which is a well-studied and reliable metric in the literature, and sample quality to measure the performance.

### 3.1 INCEPTION SCORES FOR CIFAR-10 USING DCGAN ARCHITECTURE

DCGAN is a family of architectures designed to perform well with the vanilla training procedure. They are ubiquitous in the GAN literature owing to the instability of vanilla GAN in general settings. We use this architecture to model CIFAR-10 and compare against vanilla GAN, WGAN and WGAN-GP. As WGANs need 5 discriminator iterations for every generator iteration, comparing the modeling performance can be tricky. To address this, we report two scores for vanilla GAN and DRAGAN - one using the same number of generator iterations as WGANs and one using the same number of discriminator iterations. The results are shown in Figure 5 and samples are included in the appendix (Figure 8). Notice that DRAGAN beats WGAN variants in both the configurations, while vanilla GAN is only slightly better. A key point to note here is that our algorithm is fast compared to WGANs, so in practice, the performance will be closer to the DRAGAN$^d$ case. In the next section, we will show that if we move away from this specific architecture family, vanilla GAN training can become highly unstable and that DRAGAN penalty mitigates this issue.

### 3.2 MEASURING STABILITY AND PERFORMANCE ACROSS ARCHITECTURES

Ideally, we would want our training procedure to perform well in a stable fashion across a variety of architectures (other than DCGANs). Similar to Arjovsky et al. (2017) and Gulrajani et al. (2017), we remove the stabilizing components of DCGAN architecture and demonstrate improved stability & modeling performance compared to vanilla GAN training (see appendix section 5.2.3). However, this is a small set of architectures and it is not clear if there is an improvement in general.

To address this, we introduce a metric termed the *BogoNet* score to compare the stability & performance of different GAN training procedures. The basic idea is to choose random architectures for players $G$ and $D$ independently, and evaluate the performance of different algorithms in the resulting games. A good algorithm should achieve stable performance without failing to learn or resulting in mode collapse, despite the potentially imbalanced architectures. In our experiment, each player is assigned a network from a diverse pool of architectures belonging to three different families (MLP, ResNet, DCGAN).

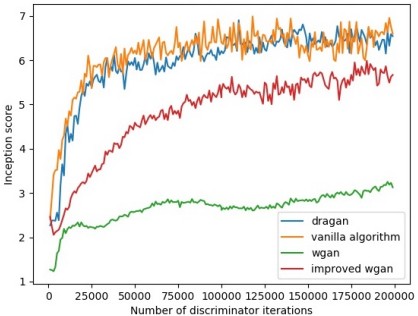

(a) Inception score plot

| Algorithm | Score |
|-----------|-------|
| WGAN | 3.25 |
| WGAN-GP | 5.99 |
| DRAGAN$^g$ | 6.11 |
| DRAGAN$^d$ | 6.90 |
| Vanilla GAN$^g$ | 6.3 |
| Vanilla GAN$^d$ | 6.99 |

(b) Inception scores

Figure 5: Comparison of modeling performance on CIFAR10

Table 1: Summary of inception score statistics across 100 architectures

| Algorithm | Final score | | Area under curve | | Qual. score |
|-----------|------|------|--------|--------|-------|
| | Mean | Std | Mean | Std | Total |
| Vanilla GAN | 2.91 | 1.44 | 277.72 | 126.09 | 92.5 |
| DRAGAN | **3.70** | 1.71 | **312.15** | 135.35 | **157.5** |
| WGAN-GP | 3.49 | 1.30 | 300.09 | 100.96 | - |

To demonstrate that our algorithm performs better compared to vanilla GAN training and WGAN-GP, we created 100 such instances of hard games. Each instance is trained using these algorithms on CIFAR-10 (under similar conditions for a fixed number of generator iterations, which gives a slight advantage to WGAN-GP) and we plot how inception score changes over time. For each algorithm, we calculated the average of final inception scores and area under the curve (AUC) over all 100 instances. The results are shown in Table 1. Notice that we beat the other algorithms in both metrics, which indicates some improvement in stability and modeling performance.

Further, we perform some qualitative analysis to verify that BogoNet score indeed captures the improvements in stability. We create another set of 50 hard architectures and compare DRAGAN against vanilla GAN training. Each instance is allotted 5 points and we split this bounty between the two algorithms depending on their performance. If both perform well or perform poorly, they get 2.5 points each, so that we nullify the effect of such non-differentiating architectures. However, if one algorithm achieves stable performance compared to the other (in terms of failure to learn or mode collapses), we assign it higher portions of the bounty. Results were judged by two of the authors in a blind manner: The curves were shown side-by-side with the choice of algorithm for each side being randomized and unlabeled. The vanilla GAN received an average score of 92.5 while our algorithm achieved an average score of 157.5 and this correlates with BogoNet score from earlier. See appendix section 5.3 for some additional details regarding this experiment.

### 3.3 STABILITY USING DIFFERENT OBJECTIVE FUNCTIONS

Our algorithm improves stability across a variety of objective functions and we demonstrate this using the following experiment. Nowozin et al. (2016) show that we can interpret GAN training as minimizing various $f$-divergences when an appropriate game objective function is used. We show experiments using the objective functions developed for Forward KL, Reverse KL, Pearson $\chi^2$, Squared Hellinger, and Total Variation divergence minimization. We use a hard architecture from the previous subsection to demonstrate the improvements in stability. Our algorithm is stable in all cases except for the total variation case, while the vanilla algorithm failed in all the cases (see Figure 6 for two examples and Figure 15 in appendix for all five). Thus, practitioners can now choose their game objective from a larger set of functions and use DRAGAN (unlike WGANs which requires a specific objective function).

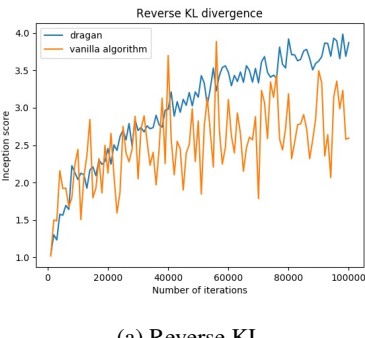
(a) Reverse KL

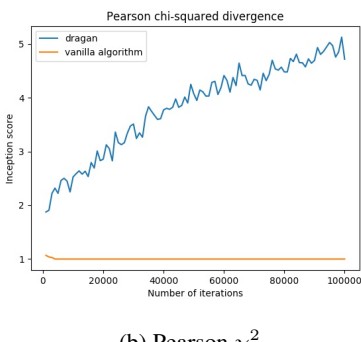
(b) Pearson $\chi^2$

Figure 6: Inception score plots for two divergence measures, demonstrating superior stability for our algorithm.

## 4 CONCLUSIONS

In this paper, we propose to study GAN training process as regret minimization, which is in contrast to the popular view that there is consistent minimization of a divergence between real and generated distributions. We analyze the convergence of GAN training from this new point of view and hypothesize that mode collapse occurs due to the existence of undesirable local equilibria. A simple observation is made about how the mode collapse situation often exhibits sharp gradients of the discriminator function around some real data points. This characterization partly explains the workings of previously proposed WGAN and gradient penalties, and motivates our novel penalty scheme. We show evidence of improved stability using DRAGAN and the resulting improvements in modeling performance across a variety of settings. We leave it to future works to explore our ideas in more depth and come up with improved training algorithms.

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

# 5 APPENDIX

## 5.1 SAMPLES AND LATENT SPACE WALKS

In this section, we provide samples from an additional experiment run on CelebA dataset (Figure 7). The samples from the experiment in section 3.1 are shown in Figure 8. Further, Radford et al. (2015) suggest that walking on the manifold learned by the generator can expose signs of memorization. We use DCGAN architecture to model MNIST and CelebA datasets using DRAGAN penalty, and the latent space walks of the learned models are shown in Figure 9 and Figure 10. The results demonstrate that the generator is indeed learning smooth transitions between different images, when our algorithm is used.

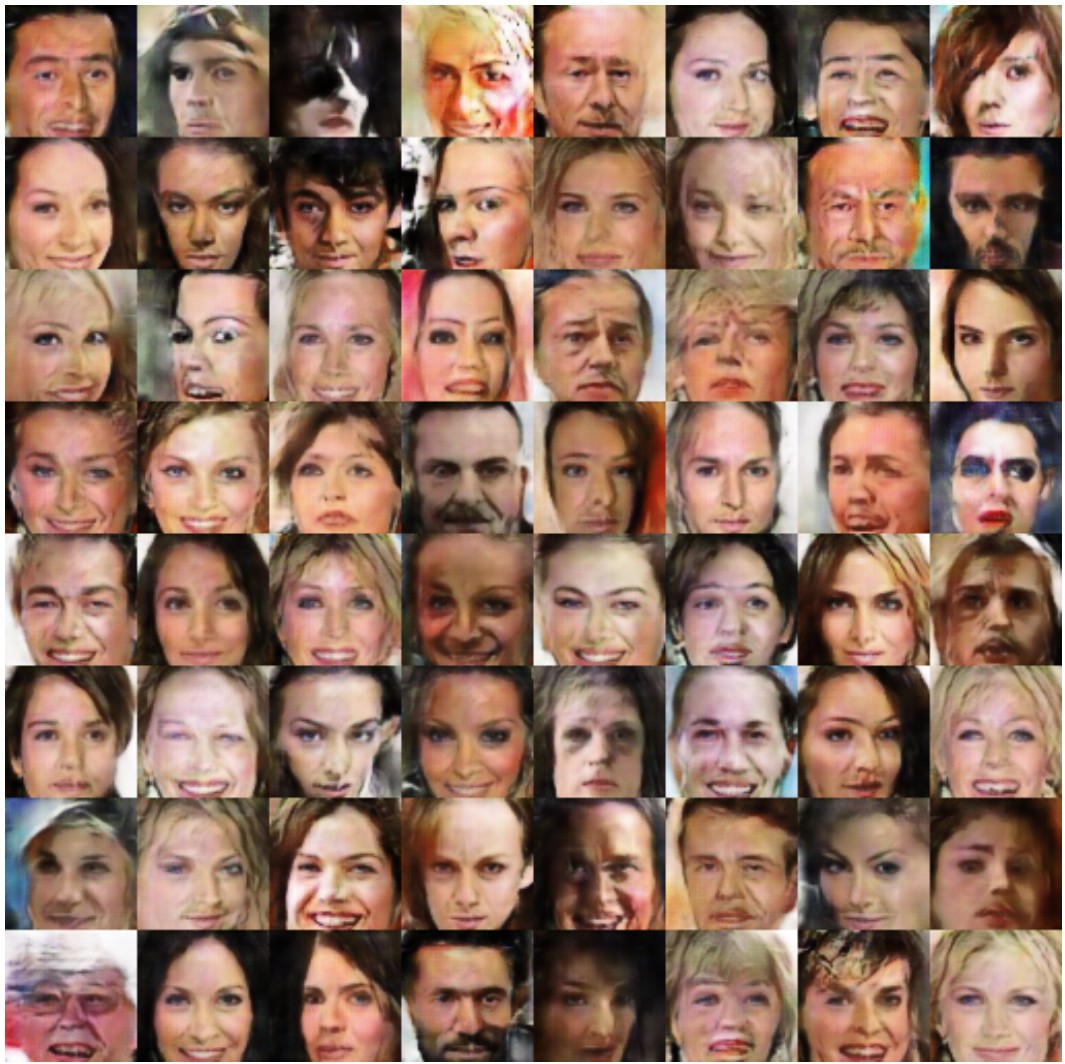

Figure 7: Modeling CelebA with DRAGAN using DCGAN architecture.

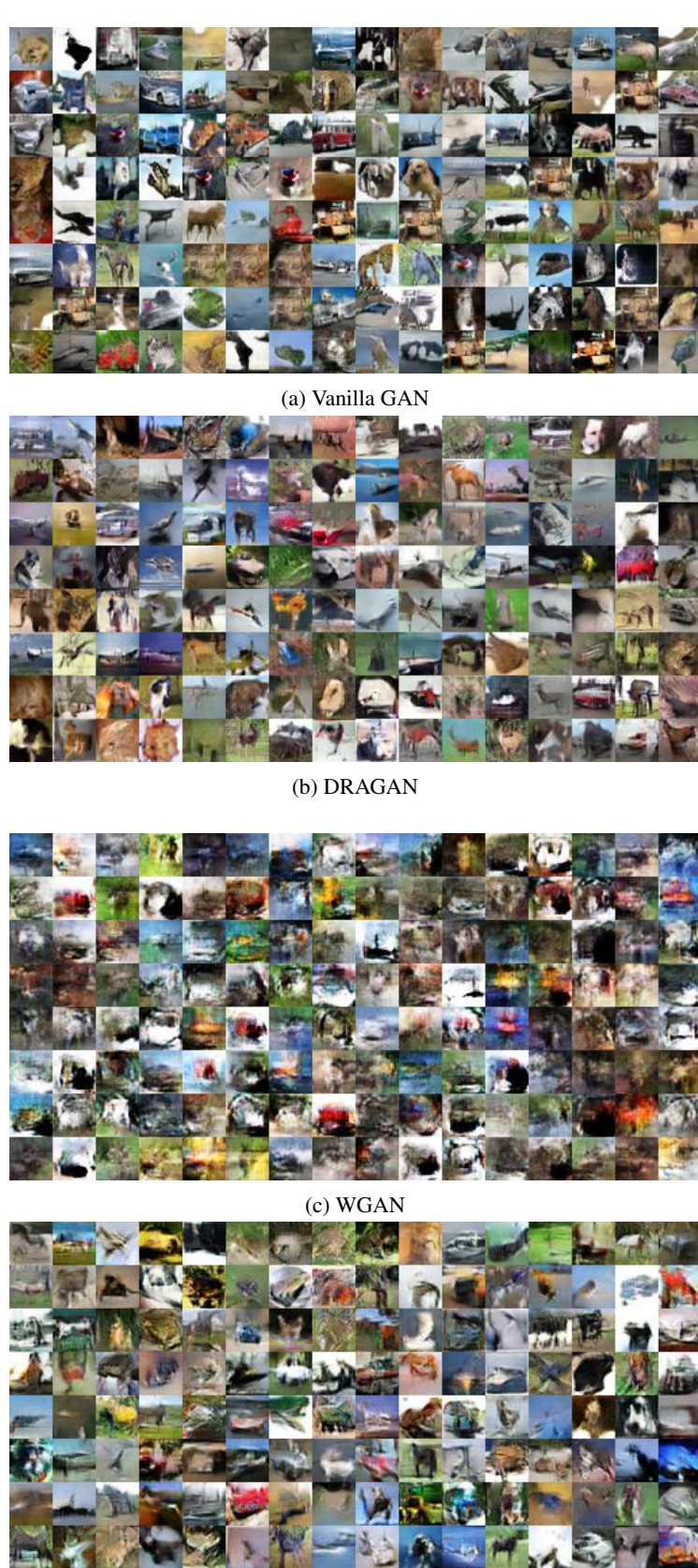

(a) Vanilla GAN

(b) DRAGAN

(c) WGAN

(d) WGAN-GP

Figure 8: Modeling CIFAR-10 using DCGAN architecture.

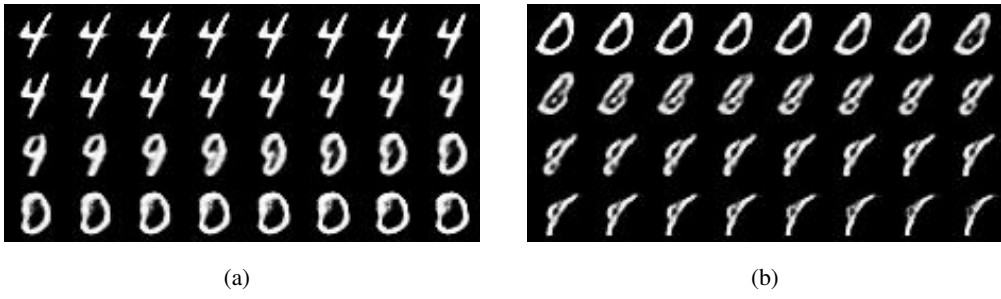

| (a) | (b) |

Figure 9: Latent space walk of the model learned on MNIST using DRAGAN

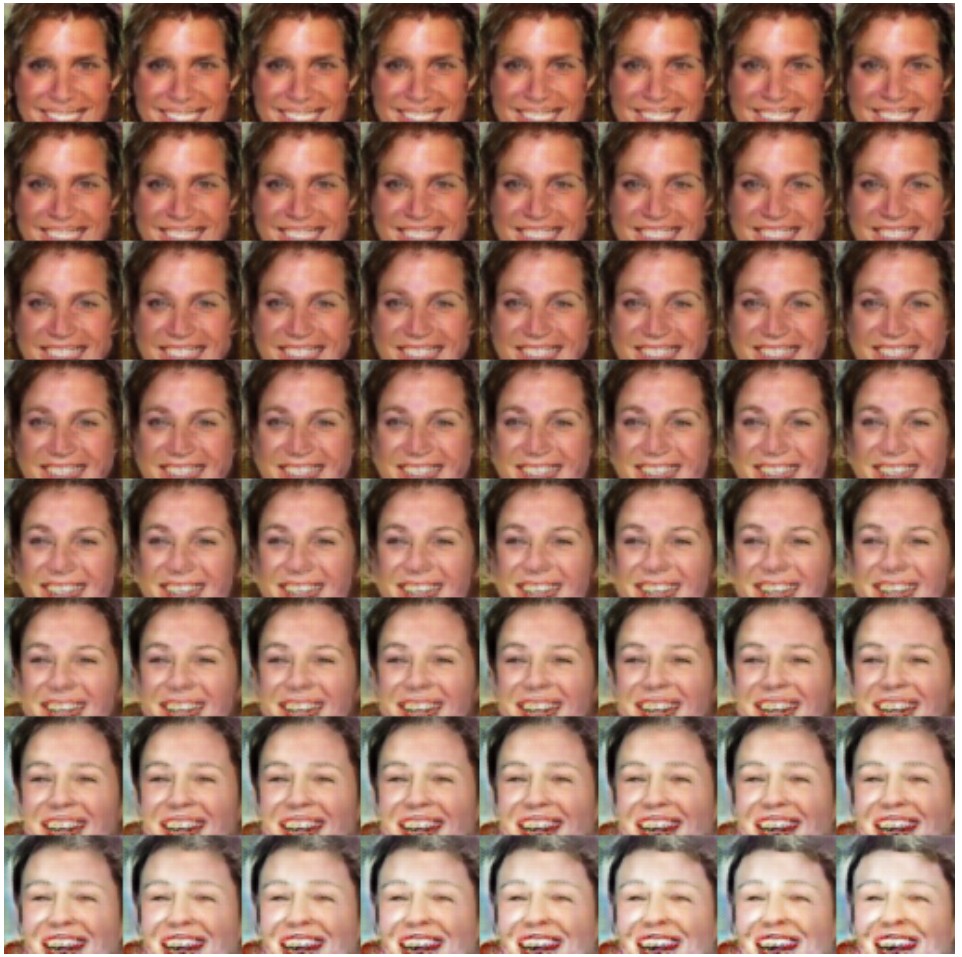

Figure 10: Latent space walk of the model learned on CelebA using DRAGAN

## 5.2 ADDITIONAL EXPERIMENTS

### 5.2.1 ONE HIDDEN LAYER NETWORK TO MODEL MNIST

We design a simple experiment where $G$ and $D$ are both fully connected networks with just one hidden layer. Vanilla GAN performs poorly even in this simple case and we observe severe mode collapses. In contrast, our algorithm is stable throughout and obtains decent quality samples despite the constrained setup.

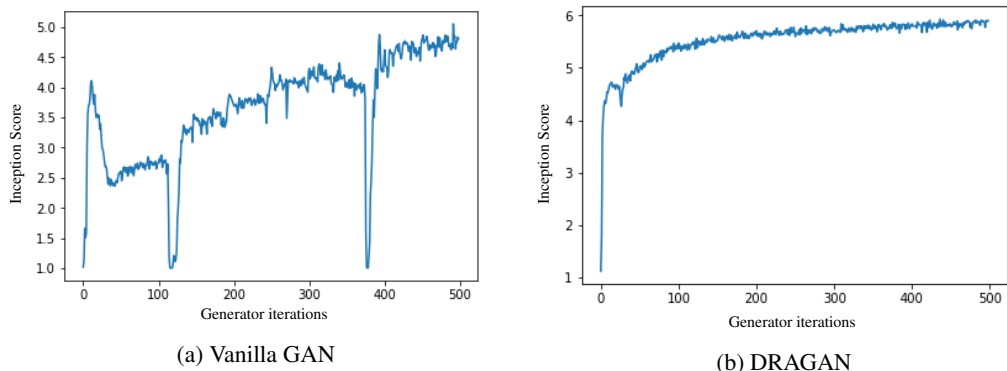

(a) Vanilla GAN

(b) DRAGAN

Figure 11: One hidden layer network to model MNIST - Inception score plots

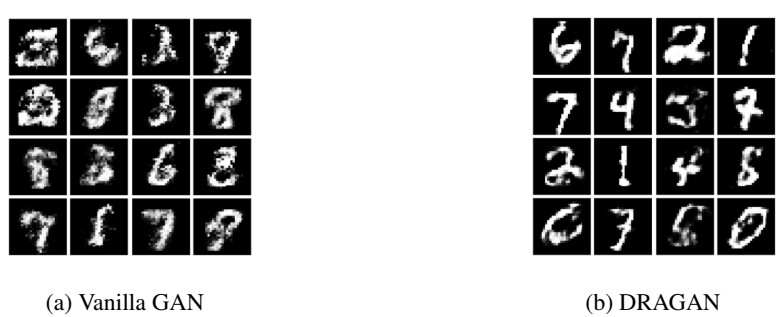

(a) Vanilla GAN

(b) DRAGAN

Figure 12: One hidden layer network to model MNIST - Samples

### 5.2.2 8-GAUSSIANS EXPERIMENT

We analyze the performance of WGAN-GP and DRAGAN on the 8-Gaussians dataset. As it can be seen in Figure 13, both of them approximately converge to the real distribution but notice that in the case of WGAN-GP, $D_\theta(x)$ seems overly constrained in the data space. In contrast, DRAGAN's discriminator is more flexible.

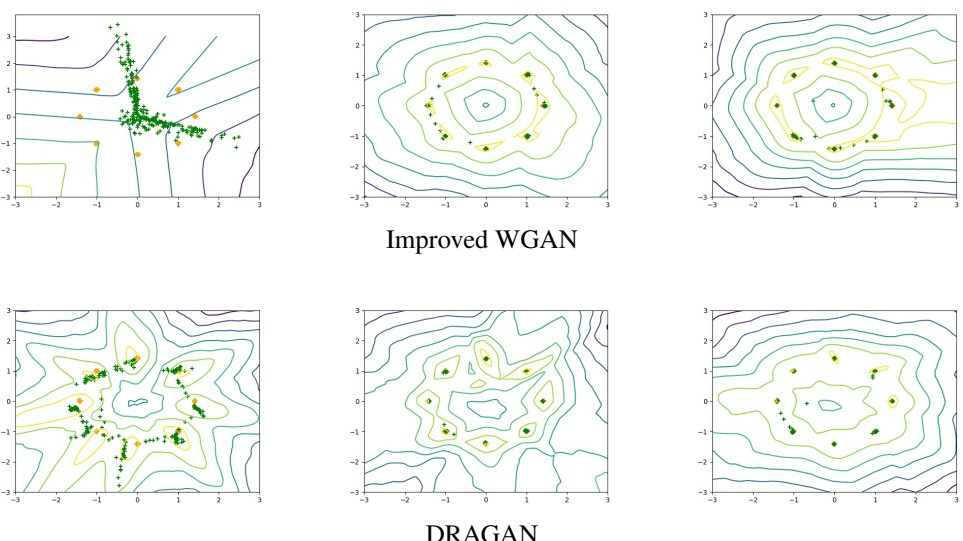

Improved WGAN

DRAGAN

Figure 13: Comparing the performance of WGAN-GP and DRAGAN on the 8-Gaussians dataset. Orange is real samples, green is generated samples. The level sets of $D_\theta(x)$ are shown in the background, with yellow as high and purple as low.

### 5.2.3 STABILITY ACROSS DCGAN ARCHITECTURE VARIATIONS

DCGAN architecture has been designed following specific guidelines to make it stable (Radford et al., 2015). We restate the suggested rules here.

1. Use all-convolutional networks which learn their own spatial downsampling (discriminator) or upsampling (generator)

2. Remove fully connected hidden layers for deeper architectures

3. Use batch normalization in both the generator and the discriminator

4. Use ReLU activation in the generator for all layers except the output layer, which uses $tanh$

5. Use LeakyReLU activation in the discriminator for all layers

We show below that such constraints can be relaxed when using our algorithm and still maintain training stability. Below, we present a series of experiments in which we remove different stabilizing components from the DCGAN architecture and analyze the performance of our algorithm. Specifically, we choose the following four architectures which are difficult to train (in each case, we start with base DCGAN architecture and apply the changes) -

- No BN and a constant number of filters in the generator

- 4-layer 512-dim ReLU MLP generator

- $tanh$ nonlinearities everywhere

- $tanh$ nonlinearity in the generator and 4-layer 512-dim LeakyReLU MLP discriminator

Notice that, in each case, our algorithm is stable while the vanilla GAN training fails. A similar approach is used to demonstrate the stability of training procedures in Arjovsky et al. (2017) and Gulrajani et al. (2017).

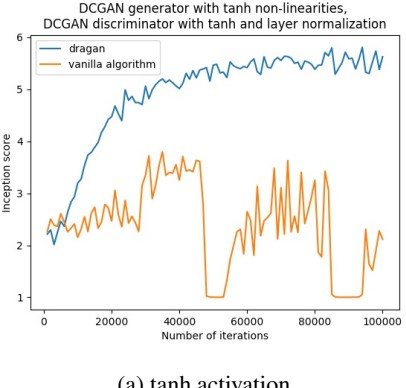

(a) tanh activation

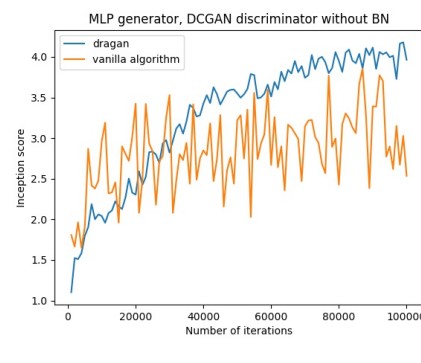

(b) FC generator

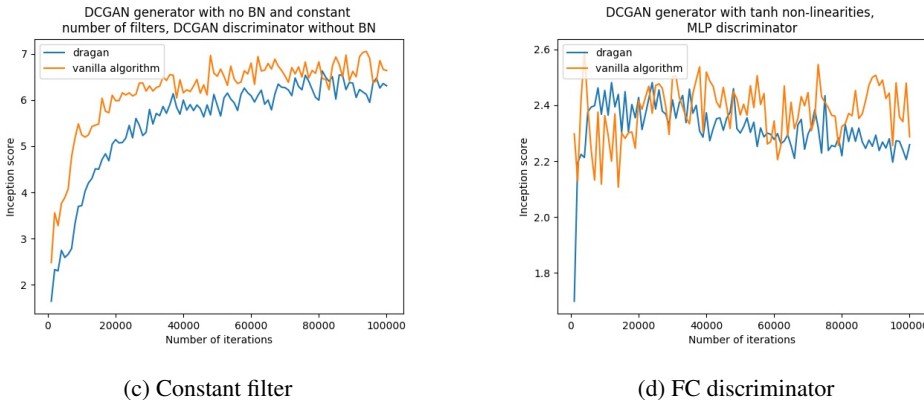

(c) Constant filter

(d) FC discriminator

Figure 14: Comparing performance of DRAGAN and Vanilla GAN training in the hard variations of DCGAN architecture.

### 5.2.4 STABILITY ACROSS OBJECTIVE FUNCTIONS

Due to space limitations, we only showed plots for two cases in section 3.3. Below we show the results for all five cases.

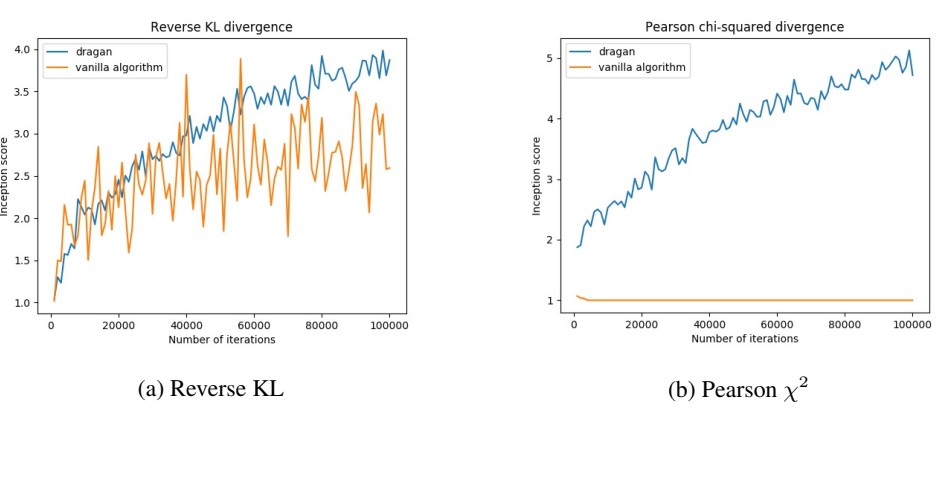

(a) Reverse KL

(b) Pearson $\chi^2$

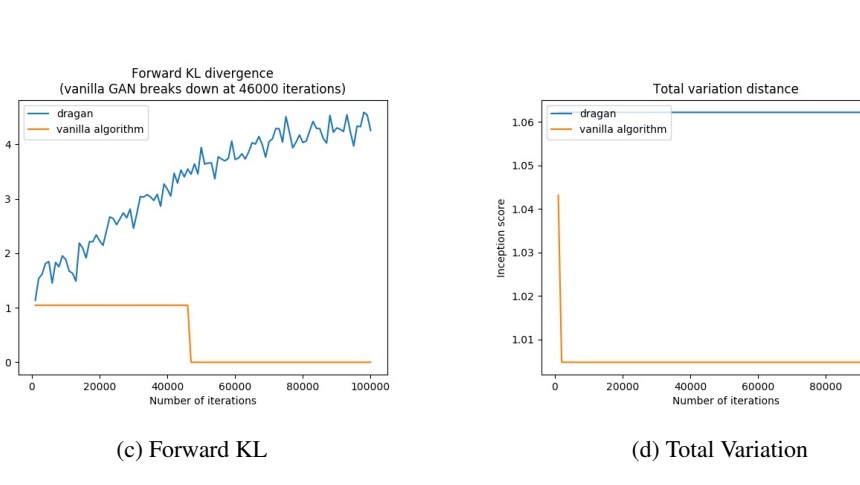

(c) Forward KL

(d) Total Variation

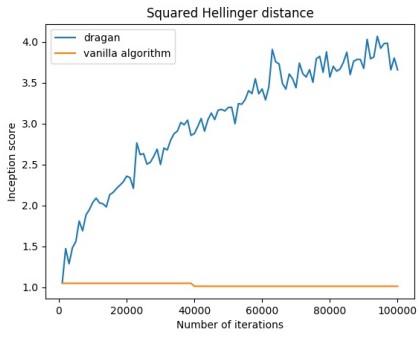

(e) Squared Hellinger

Figure 15: Comparing performance of DRAGAN and Vanilla GAN training using different objective functions.

## 5.3   BOGONET DETAILS

We used three families of architectures with probabilities - DCGAN (0.6), ResNet (0.2), MLP (0.2). Next, we further parameterized each family to create additional variation. For instance, the DCGAN family can result in networks with or without batch normalization, have LeakyReLU or Tanh non-linearities. The number and width of filters, latent space dimensionality are some other possible variations in our experiment. Similarly, the number of layers and hidden units in each layer for MLP are chosen randomly. For ResNets, we chose their depth randomly. This creates a set of hard games which test the stability of a given training algorithm.

We showed qualitative analysis of the inception score plots in section 3.2 to verify that BogoNet score indeed captures the improvements in stability. Below, we show some examples of how the bounty splits were done. The plots in Figure 14 were scored as (averages are shown in DRAGAN, Vanilla GAN order):

A - (5, 0), B - (3.5, 1.5), C – (2.25, 2.75), D – (2, 3)

