# OpenReview forum: "On Convergence and Stability of GANs"
_ICLR.cc/2018/Conference — Reject_

### Official Review · AnonReviewer2 · 2017-11-10
**A simple regularization term for training GANs is introduced, with good numerical performance.**

**Rating:** 5
**Confidence:** 2

**Review:**

Summary
========
The authors present a new regularization term, inspired from game theory, which encourages the discriminator's gradient to have a norm equal to one. This leads to reduce the number of local minima, so that the behavior of the optimization scheme gets closer to the optimization of a zero-sum games with convex-concave functions.


Clarity
======
Overall, the paper is clear and well-written. However, the authors should motivate better the regularization introduced in  section 2.3.


Originality
=========
The idea is novel and interesting. In addition, it is easy to implement it for any GANs since it requires only an additional regularization term. Moreover, the numerical experiments are in favor of the proposed method.


Comments
=========
- Why should the norm of the gradient should to be equal to 1 and not another value? Is this possible to improve the performance if we put an additional hyper-parameter instead?

- Are the performances greatly impacted by other value of lambda and c (the suggested parameter values are lambda = c = 10)?

- As mentioned in the paper, the regularization affects the modeling performance. Maybe the authors should add a comparison between different regularization parameters to illustrate the real impact of lambda and c on the performance.

- GANs performance is usually worse on very big dataset such as Imagenet. Does this regularization trick makes their performance better?



Post-rebuttal comments
---------------------------------

I modified my review score, according to the problems raised by Reviewer 1 and 3. Despite the idea looks pretty simple and present some advantages, the authors should go deeper in the analysis, especially because the idea is not so novel.

---

> ### Author Response · Authors · 2017-12-07
> **Thanks for the feedback and useful comments**
>
> - A small correction in your summary. Our penalty scheme helps avoid bad local equilibria and the convex-concave case, while being simple, is quite different from the non-convex case.
>
> - We changed section 2.3 to rigorously argue that regret minimization converges to (potentially bad) local equilibria, added a new section 2.4 to characterize what these 'mode collapse' equilibria look like (D has large gradients around real samples in this case) and demonstrate that they can be averted using gradient constraints, through new toy experiments. This provides good intuition and a strong motivation for the introduction of DRAGAN scheme. We urge you to take another look at sections 2.3, 2.4.
>
> - In the updated revision, we correct this arbitrary choice and use 'k', which should be something small. Basically, we observe that 'mode collapse' equilibria exhibit sharp gradients of the discriminator function around real samples. So, we regularize so as to keep these gradients small. We apologize for not making it clear earlier.
>
> - You make an excellent point that by tuning 'k'/'c'/'lambda', it could be possible to get better performance but our aim here was just to demonstrate the effectiveness of our method. My intuition is that optimal configuration will depend on data domain, architecture and hence, its beyond the scope of our paper. But, this is an important topic for possibly a future work.
>
> - We only explore the performance of our penalty on MNIST, CIFAR-10 and CelebA, like most papers in this direction. I think the performance on ImageNet depends heavily on the architecture but we did not explore this aspect in our paper. It is an interesting topic to compare various methods on bigger datasets, maybe using ResNets.

---

### Official Review · AnonReviewer1 · 2017-11-25
**Rather incremental work, I doubt the scientific contribution is significant**

**Rating:** 4
**Confidence:** 5

**Review:**

This paper addresses the well-known stability problem encountered when training GANs. As many other papers, they suggest adding a regularization penalty on the discriminator which penalizes the gradient with respect to the data, effectively linearizing the data manifold.

Relevance: Although I think some of the empirical results provided in the paper are interesting, I doubt the scientific contribution of this paper is significant. First of all, the penalty the author suggest is the same as the one suggest by Gulrajani for Wasserstein GAN (there the motivation behind this penalty comes from the optimal transport plan). In this paper, the author apply the same penalty to the GAN objective with the alternative update rule which is also a lower-bound for the Wasserstein distance.

Justification: The authors justify the choice of their regularization saying it linearizes the objective along the data manifold and claim it reduces the number of non-optimal fixed points. This might be true in the data space but the GAN objective is optimized over the parameter space and it is therefore not clear to me their argument hold w.r.t to the network parameters. Can you please comment on this?

Regularizing the generator: Can the authors motivate their choice for regularizing the discriminator only, and not the generator? Following their reasoning of linearizing the objective, the same argument should apply to the generator.

Comparison to existing work: This is not the first paper that suggests adding a regularization. Given that the theoretical aspect of the paper are rather weak, I would at least expect a comparison to existing regularization methods, e.g.
Stabilizing training of generative adversarial networks through regularization. NIPS, 2017

Choice of hyper-parameters: The authors say that the suggested value for lambda is 10. Can you comment on the choice of this parameter and how it affect the results? Have you tried  annealing lambda? This is a common procedure in optimization (see e.g. homotopy or continuation methods).

Bogonet score: I very much like the experiment where the authors select 100 different architectures to compare their method against the vanilla GAN approach. I here have 2 questions:
- Did you do a deeper examination of your results, e.g. was there some architectures for which none of the method performed well?
- Did you try to run this experiment on other datasets?

---

> ### Author Response · Authors · 2017-12-07
> **Thanks for your insightful feedback. We clarify the significance of our theory contributions.**
>
> Clarification regarding the importance of our theory sections:
>
> We admit that the clarity in our presentation was lacking (especially ties to the GAN literature) and tried to address it in the new revision. We urge you to please take another look. Specifically, our contributions are: (reflected in updated abstract and introduction):
>
> - We propose to study GAN training as regret minimization. This is a completely novel contribution. In contrast, the popular view is that there is consistent divergence minimization and this is based on the unrealistic assumption that the discriminator is playing optimally at each step and making these updates in the function space. This isn't tractable (nor) close to what happens in practice. This forms the main motivation for our paper.
>
> - We present the analysis of artificial convex-concave case based on standard results in game theory literature. More importantly, we make explicit the connection between GAN training process (alternating gradient updates) and regret minimization, along the way, in section 2.2. These are not widely known results in the GAN literature and we provide supporting references in the new revision.
>
> **The useful outcome is that this analysis yields a novel proof for the asymptotic convergence of GAN training in the non-parametric limit and it does not require the discriminator to be optimal at each step**
> The current revision reflects this message.
>
> - To explain mode collapse, we next analyze the realistic non-convex case in section 2.3 from regret minimization perspective. This is very different from the convex-concave case actually (we apologize for the confusion) and we cite the works of Hazan et.al to rigorously argue that convergence to potentially bad local equilibria happens using gradient updates (under some conditions). Please see the updated section 2.3.
>
> This leads us to the main hypothesis of our paper - that mode collapse is just an undesirable local equilibrium and it should be possible to avoid it. We apologize for not being clear earlier. The natural question now is how we can avoid these equilibria?
>
> - A new section 2.4 has been added which explains how we can avoid 'mode collapse' equilibria (this was implicit and not clear earlier). Based on empirical observations, we basically characterize 'mode collapse' equilibria with sharp gradients of the discriminator function around some real data points. This is key to fighting mode collapse and avoiding such undesirable equilibria. We provide arguments and supporting experiments for this in a new section 2.4. This was a key transition that was missing in the earlier version.
>
> - From this motivation (of keeping D's gradients small in ambient data space), we propose DRAGAN penalty scheme. In fact, our theory also explains how other gradient penalties (WGAN-GP/LS-GAN) might be mitigating mode collapse. We compare and discuss its advantages over them in section 2.5. And present the main experiments in section 3.

---

> > ### Author Response · Authors · 2017-12-07
> > **(Continued)**
> >
> > DRAGAN algorithm:
> >
> > - You are right that our penalty is applied on top of the vanilla objective. But our penalty (local penalty) is also quite different from WGAN-GP and LS-GAN (coupled penalties, both of these are shown to be very similar in our paper) as we only regularize in local regions around the real data. We dedicate the entire section 2.5 to compare/contrast these methods.
> >
> > - Further, we discuss how WGAN-GP's gradient penalty has little to do with Wasserstein duality as claimed in their paper (please see section 2.5) and in fact, this adds more credence to our theory that keeping D(x) gradients around real data to be small, is how they mitigate mode collapse.
> >
> > - The explanation for why constraining D(x)'s gradients around real data helps, is provided in section 2.4. And adding this penalty to the cost function of D and performing gradient descent w.r.t parameters, encourages the player to learn smooth functions which is what we want. This same idea has been used in Gulrajani et.al, Qi et.al as well. It would be interesting to come up with architectural design principles that inherently result in smooth discriminator functions.
> >
> > - We observed that 'mode collapse' equilibria exhibit sharp gradients of the discriminator function around real points and hence, we propose regularizing D (a gradient penalty scheme). Our work does not study mode collapse from the generator's perspective but what you propose is an excellent research direction. I think the method to achieve stability will look very different if one takes that approach since the generator's architecture is significantly different.
> >
> > - Our work mainly deals with the question of why mode collapse happens from 'regret minimization' perspective. And connect that to gradient penalties for constraining D's gradients in the data space. Our work was done prior to Roth et.al and so, we were only able to compare with WGAN-GP. But they also suggest a similar gradient penalty.
> >
> > - This is an excellent point that one should perform annealing of lamdba to get the best results using regularization schemes. However, we focus in our paper on why mode collapse happens, how we can characterize it and methods to avoid it. As long as D(x) has small gradients around real data, mode collapse can be averted. Our aim was not to get the best experimental results, and we only wanted to demonstrate the effectiveness of our scheme.
> >
> >
> > BogoNet Score:
> >
> > - We did observe architectures where both the methods performed well and cases where both of them failed. To nullify the effect of such non-differentiating architectures, our bounty model awards 2.5 points each (out of 5) in such cases.
> >
> > - This experiment was only done using the standard CIFAR-10 dataset. Due to the constraints on GPU resources available to us, we couldn't try different datasets. Especially, since we included ResNets in the experiment which take days to train. But, what you suggest is an interesting experiment as well.

---

### Official Review · AnonReviewer3 · 2017-11-25
**Lack of the main point**

**Rating:** 3
**Confidence:** 3

**Review:**

This paper contains a collection of ideas about Generative Adversarial Networks (GAN) but it is very hard for me to get the main point of this paper. I am not saying ideas are not interesting, but I think the author needs to choose the main point of the paper, and should focus on delivering in-depth studies on the main point.

1. On the game theoretic interpretations

The paper, Generative Adversarial Nets, NIPS 2014, already presented the game theoretic interpretations to GANs, so it's hard for me to think what's new in the section. Best response dynamics is not used in the conventional GAN training, because it's very hard to find the global optimal of inner minimization and outer maximization.
The convergence of online primal-dual gradient descent method in the minimax game is already well-known, but this analysis cannot be applied to the usual GAN setting because the objective is not convex-concave. I found this analysis would be very interesting if the authors can find the toy example when GAN becomes convex-concave by using different model parameterizations and/or different f-divergence, and conduct various studies on the convergence and stability on this problem.

I also found that the hypothesis on the model collapsing has very limited connection to the convex-concave case. It is OK to form the hypothesis and present an interesting research direction, but in order to make this as a main point of the paper, the author should provide more rigorous arguments or experimental studies instead of jumping to the hypothesis in two sentences. For example, if the authors can provide the toy example where GAN becomes convex-concave vs. non-convex-concave case, and how the loss function shape or gradient dynamics are changing, that will provide very valuable insights on the problem.

2. DRAGAN

As open commenters pointed out, I found it's difficult to find why we want to make the norm of the gradient to 1.
Why not 2? why not 1/2? Why 1 is very special?
In the WGAN paper, the gradient is clipped to a number less than 1, because it is a sufficient condition to being 1-Lipshitz, but this paper provides no justification on this number.
It's OK not to have the theoretical answers to the questions but in that case the authors should provide ablation experiments. For example, sweeping gradient norm target from 10^-3, 10^-2, 10^-1, 1.0, 10.0, etc and their impact on the performance.
Also scheduling regularization parameter like reducing the size of lambda exponentially would be interesting as well.
Most of those studies won't be necessary if the theory is sound. However, since this paper does not provide a justification on the magic number "1", I think it's better to include some form of ablation studies.

Note that the item 1 and item 2 are not strongly related to each other, and can be two separate papers. I recommend to choose one direction and provide in-depth study on one topic. Currently, this paper tries to present interesting ideas without very deep investigations, and I cannot recommend this paper to be published.

---

> ### Author Response · Authors · 2017-12-07
> **Thanks for the excellent review, it was very helpful**
>
> We found your review to be extremely helpful to understand where our paper was lacking. Our paper did have multiple new ideas and the presentation wasn't always clear (ties to GAN literature were missing). Thanks to your feedback, we chose the most important strand and strengthened it in the current revision using additional theoretical arguments and targeted experiments. The core content is still the same but the presentation has been changed significantly for improved clarity. We urge you to take another look.
>
> Specifically, the main points now read as (reflected in updated abstract and introduction):
> - We propose to study GAN training as regret minimization (this is a novel view), which is in contrast to the popular view that there is consistent divergence minimization.  More about this below.
> - We provide a novel proof for the asymptotic convergence of GAN training in the non-parametric limit and it does not require the discriminator to be optimal at each step.
> - Regret minimization (AGD) in non-convex games converges to potentially bad local equilibria, under some conditions and we hypothesize mode collapse to be resulting from this. Please see updated section 2.3, where we added theoretical arguments to support this. Next question is how we can avoid these equilibria?
> - We characterize mode collapse equilibria with sharp gradients of the discriminator function around some real data points. We provide toy experiments and arguments in a new section 2.4 to support this. We apologize for missing this key transition earlier.
> - From this motivation, we propose DRAGAN penalty scheme. We compare and discuss advantages over WGAN-GP, LS-GAN in section 2.5. And present main experiments in section 3.
>
> Theory sections:
>
> 1. We should have called section 2.1 as background since its reviewing original GAN paper's formulation and we apologize for not making it clear.
>
> 2. However, studying GAN training dynamics as regret minimization is a completely novel contribution. And the popular divergence minimization hypothesis stems from D using best-response algorithm in the function space. This isn't tractable as you mention, nor close to what happens in practice. In fact, this is the main motivation for our paper.
>
> 3. We agree with you that the content in section 2.3 is well-known in game theory literature and moreover, the convex-concave is artificial. Our goal here was to simply review these results, introduce formal notions along the way, which we use in later sections and most importantly, make explicit the connection between GAN training (alternating gradient updates) and regret minimization. None of this is widely known in GAN literature and we support this claim with references as recent as 2017.
> **The useful outcome is that the analysis yields a novel proof for the asymptotic convergence of GAN training in the non-parametric limit and it does not require the discriminator to be optimal at each step**
> The current revision reflects this message.
>
> 4. To explain mode collapse, we analyze the realistic non-convex case in section 2.3. This is very different from convex-concave case actually (we apologize for the confusion) and we cite the works of Hazan et.al to rigorously argue that convergence to local equilibria can be expected using regret minimization or OGD. This leads us to the main hypothesis of our paper - that mode collapse is just an undesirable local equilibrium and it should be possible to avoid it. We apologize for not being clear earlier.
>
> 5. A new section 2.4 has been added which explains how we can avoid 'mode collapse' equilibria. Based on empirical observations, we characterize mode collapse situation with sharp gradients of the discriminator function around some real data points. This is key to fighting mode collapse or avoiding such undesirable equilibria. We provide arguments and supporting experiments for this. From this motivation, DRAGAN penalty scheme is introduced. This was a key transition that was missing in the earlier version.
>
> DRAGAN algorithm:
>
> 1. From the strengthened theory sections, it should be clear that as long as D(x) has small gradients around real data, mode collapse can be mitigated. We removed the arbitrary '1' in our scheme and used a generic 'k' (some small constant). We apologize for the jump earlier.
> 2. The key idea is keeping D(x) gradients small and this stems from our observation that 'mode collapse' equilibria can be characterized by large gradients of D in the data space. In fact, this partly explains why WGAN-GP and LS-GAN improve stability, despite being motivated by reasoning (divergence minimization hypothesis) that is based on unrealistic assumptions. We urge you to take another look at sections 2.3 and 2.4.

---

### Public Comment · (anonymous) · 2017-11-02
**Intuition of the regularization**

 This paper is very well written. It is very easy to follow. It takes me less than 10 mins to read the whole paper. I have a question on the regularization term. Why does it require the norm of gradient to be close to 1? When D(x) approaches the optimal, the gradient should be zero, then this additional regularization term would make the training unstable or not converge to the optimal?

---

> ### Author Response · Authors · 2017-11-02
> **Thanks for your feedback!**
>
> 1. As we explain at the end of section 2.4, one can constrain D in multiple ways and still improve stability. It should be clear from our paper that stability requires trading off modeling performance as flexible models come with game-theoretic problems.
>
> We chose this specific form using our intuitions so as to have the least negative impact on modeling performance. Hence, we only apply constraints near real samples unlike other approaches. Next, we wanted that D be "smooth" in x-space to help the generator learn better and change gradually w.r.t theta so that game dynamics improves (see why FTRL works in previous section for intuitions regarding this).
>
> Let's see why our constraint achieves both of these. It is reasonable to expect that almost any small pixel-wise perturbation will make a given image less realistic. So we want that D(x) and D(x') to be different and to somewhat depend on how far x and x' themselves are. Thus, gradient should be greater than zero or the generator cannot learn to tell real images and the noise apart. Of course, you can play around with that parameter but we found that it doesn't matter much (atleast for stability). The gradual change in D w.r.t theta happens as these local perturbations act as auxiliary data points holding D down (in some sense) to prevent rapid changes.
>
> 2. To answer your second question, please look back to our section 2.3 where we outline the possibilities for theta and phi in non-convex settings. They can:
>
> -> Converge to an equilibrium (can be local)
> -> Cycle s.t averages converge
> -> Don't converge at all
>
> If D(x) converges to optimal, notice that this means we are "almost" in a local equilibria. The cost function for G is now fixed and he will perform SGD updates until reaching a local minima mostly. This is usual deep learning and so, there's no question of instability due to the game! Whether this result is optimal depends on well-posedness of the game. This is where our paper comes in :) As we explain in our conclusion, the dynamics of GANs are not understood in the right perspective yet. Thinking of them as consistently estimating and minimizing JS-divergence or Wasserstein distance is not appropriate.

---

> > ### Public Comment · (anonymous) · 2017-11-02
> > **follow up question**
> >
> >
> > There is one thing that is puzzling me. Suppose D* is the optimal to the original GAN without the regularization term. The gradient of D* with respect to x should be zero. Suppose we let D' be the optimal point for the new objective function with the regularization term. If the regularization term dominates, the converging point D' would have a gradient whose norm is close to 1. In this case, D' may be very different from the original D*, and the generated distribution pg may be very different from data distribution. My question is why do we require the norm of the gradient to be close to 1, instead of any other number? For example, if the regularization requires the norm of the gradient to be close to some small constant, would the converging D' be more close to D*?

---

> > > ### Author Response · Authors · 2017-11-02
> > > **D* won't have zero gradient w.r.t X in general.**
> > >
> > > It is incorrect to assume that D* will have zero gradient w.r.t X, which means that D* will be a constant function and there's no reason to believe this will happen. However, w.r.t theta, gradient will be zero as its optimal.
> > >
> > > Now, let's talk about D* vs D'. You are right that D' can be worse than D*, however, as we discuss in the paper, getting to D* is a perilous journey fraught with local equilibria/instabilities. But, add the regularization term and you get D' which we show isn't that worse off. But now, you get significantly improved stability!
> > >
> > > The generated distribution isn't close to P_real in both the cases, atleast we have no reason to believe so, expect using visual inspection (which can be a slippery slope). But w.r.t metrics we have (inception score, visual inspection), D' and D* are almost the same in our experiments.
> > >
> > > The explanation could be that constraining to have norm-k gradients is actually reasonable (perturbed images should get strictly smaller probability than actual images) and P_real itself satisfies this condition. So, with large enough data D' -> P_real just as D*-> P_real, except that with DRAGAN, we are more likely to reach there due to stability.

---

> > > > ### Public Comment · (anonymous) · 2017-11-02
> > > > **Why D* does not have zero gradient w.r.t. X**
> > > >
> > > > Dear authors,
> > > > I am new to this area, so I have quite a few questions. I understand that the experiments show that adding this norm regularization makes the performance good and stable. The paper is very well written and the results are actually very impressive. I just want to step back and seek for a theoretical explanation of why this regularization makes sense.
> > > >
> > > > 1. If we look at the original minimax problem of GAN, the optimal D* is indeed D*(x) = 1/2. Why is it not correct to assume D* has zero gradient w.r.t. X?
> > > >
> > > > 2.  Let us first step back and assume the minimax problem is convex/concave in theta_d/theta_g. When we design the algorithm, we should find one that at least works for the ideal case, and then think about how to make it robust for the general case, when the problem is not convex/concave in theta_d/theta_g. I understand that adding the norm regularization in the data support would definitely alleviate mode collapse, because it artificially adds gradients to encourage the generator to generate the data samples. My question is  why should we restrict the norm to be close to "1" instead of some other small numbers (or some vanishing numbers), which would make your D' hopefully most closely to D*?

---

> > > > > ### Author Response · Authors · 2017-11-02
> > > > > **Misleading to use the intuition that D* actually represents the ratio of densities in practice.**
> > > > >
> > > > > 1. Goodfellow et.al show D*(x)=1/2 as we converge to P_real, when we have infinite data and large (maybe infinite) capacity networks. This isn't a realistic setting and it can be misleading to use the intuition that D* actually represents the ratio of densities in practice.
> > > > >
> > > > > So, D*(x) need not have zero gradient w.r.t X, in general.
> > > > >
> > > > > 2. Now, the generator learns from D in GAN framework. All G cares about is getting high scores and all D cares about is providing high scores to only real samples! What happens at noise doesn't matter as long as they get strictly lower scores.
> > > > >
> > > > > If you use vanishing numbers, then you encourage the generator to learn noise and I think you are suggesting to slowly remove the regularization, which is a good idea in the limit case.

---

> > > > > > ### Public Comment · (anonymous) · 2017-11-02
> > > > > > **any norm regularization works?**
> > > > > >
> > > > > >  I see. Actually, quite a few people consider D(x) as the density ration estimation:
> > > > > > M Uehara, "Generative Adversarial Nets from a Density Ratio Estimation Perspective"
> > > > > > B. Poole et al. , "Improved generator objectives for GANs".
> > > > > > And the analysis for the case with finite sample data is also aligned with this perspective.  See Arora et al, "Generalization and Equilibrium in Generative Adversarial Nets (GANs)"
> > > > > > Probably you are right, in actual training, it may be misleading to consider from this perspective.
> > > > > >
> > > > > > Following the logic, does it imply that the any gradient norm regularization should work? So the constant "1" is chosen heuristically based on experiment results? Is there any general guideline to choose the norm regularization constant, should it be always "1"?

---

> > > > > > > ### Author Response · Authors · 2017-11-02
> > > > > > > **Arora et al is a great paper which supports what I said...how it's misleading to apply asymptotic intuitions in practice.**
> > > > > > >
> > > > > > > Arora et al's is a great paper which supports what I said...how it's misleading to apply asymptotic intuitions in practice.
> > > > > > >
> > > > > > > From a game-theoretic perspective, yes, any form of gradient norm regularization should improve stability. Our goal was to demonstrate this idea and foster research in this direction.
> > > > > > >
> > > > > > > We didn't explore all possibilities or claim to have the best answer here, this is beyond the scope of our work. In fact, we didn't even explore all numerical possibilities to optimize for performance! And yet we beat the state-of-the-art wgan-gp. Hopefully, practitioners will build off our work and develop better algorithms.

---

> > > > > > > > ### Public Comment · (anonymous) · 2017-11-02
> > > > > > > > **Look forward to more theoretical insights in future!**
> > > > > > > >
> > > > > > > >  Thanks a lot for your explanations!
> > > > > > > >
> > > > > > > > Actually if you look at the WGAN-GP paper, they have a strong theoretical support for adding the norm regularization term, because ". A differentiable function is 1-Lipschtiz if and only if it has gradients with norm at most 1 everywhere".
> > > > > > > >
> > > > > > > > For your paper, there is no doubt that the regularization brings improvement in the experiments. The paper is very well written. Besides the good experiment performance, I am looking for similar theoretical insights of adding this regularization for other variants of GANs. Instead of knowing that it works, I am more interested in knowing why it works, because obtaining the theoretical insights will lead to more effective algorithms that can be generalized. Hopefully, we will obtain more theoretical insights from future works, if more practitioners build more algorithms on this.

---

> > > > > > > > > ### Author Response · Authors · 2017-11-02
> > > > > > > > > **WGAN-GP is a great heuristic but it is just motivated/inspired by KR duality**
> > > > > > > > >
> > > > > > > > > 1. "A differentiable function is 1-Lipschtiz if and only if it has gradients with norm at most 1 everywhere", but WGAN-GP doesnt do this.
> > > > > > > > >
> > > > > > > > > 2. Read our section 2.4 where we show WGAN-GP has little to do with Wasserstein duality. In fact, our game-theoretic arguments could be the basis for why it works to some extent.
> > > > > > > > >
> > > > > > > > > 3. Most of other such GAN variants come up with techniques by applying asymptotic arguments or sometimes those that don't hold in practice! Our paper is trying to counter that and we just follow the game.
> > > > > > > > >
> > > > > > > > > Our section 2.3 is the starting point for theory you are looking for. However, I suggest to carefully think about assumptions made in the development of each algorithm

---

> > > > > > > > > > ### Public Comment · (anonymous) · 2017-11-02
> > > > > > > > > > **Thanks for your clarification**
> > > > > > > > > >
> > > > > > > > > > I understand that your analysis suggests that we should add the norm regularization in the local areas around real examples. This is clearly an improvement over WGAN-GP. However, this norm~=1 only holds for the case of Wasserstein distance. This is also what motivates Gulrajani et al to introduce this regularization term in the objective function.
> > > > > > > > > >
> > > > > > > > > > However, for other variants of GANs based on general f-divergence, the optimal D* should not have a norm~=1. That is what puzzling me and want to seek for theoretical insights.
> > > > > > > > > >
> > > > > > > > > > I agree that in practical training of GANs. We do not have infinite data samples, the network may not have enough capacity, and the neural networks are not convex/concave over its parameters. But when we design our algorithm, should we design one that at least works for the ideal simplest case when we have the true expectation, the network has enough capacity and the global optimum points can be reached? I think we should at least guarantee the algorithm works for the ideal case and then think about how to make it robust for practice.

---

> > > > > > > > > > > ### Author Response · Authors · 2017-11-02
> > > > > > > > > > > **Wasserstein distance is if we use infinite family of 1-Lip functions**
> > > > > > > > > > >
> > > > > > > > > > > Wasserstein distance is if we use infinite family of 1-Lip (norm of gradient <=1) functions.
> > > > > > > > > > >
> > > > > > > > > > > But, wgan-gp forces norm-1 gradients between all real and fake pairs. So, there is little connection to Wasserstein duality theory here (asymptotic or otherwise).
> > > > > > > > > > >
> > > > > > > > > > > I agree that more theoretical investigation is needed in the community. But this process of using only asymptotic intuitions to develop algorithms can go wrong as there are many moving pieces in GAN framework.
> > > > > > > > > > >
> > > > > > > > > > > And your question of whether we should require our algorithm to work in limit case..yes, absolutely! But do we understand what the right notion is? Density ratio was one way to think about this as suggested by original Goodfellow's paper. But this breaks the moment infinite data assumption is relaxed. So, we are yet to find a way to nicely reason about limit case and until we have it, using such narratives is overly restrictive. In our paper, we see D as some entity that can tell real images vs everything else and hence, our penalty makes sense.

---

> > > > > > > > > > > > ### Public Comment · (anonymous) · 2017-11-02
> > > > > > > > > > > > **Thanks!**
> > > > > > > > > > > >
> > > > > > > > > > > > Thanks!

---

### Public Comment · ~Leon_Boellmann1 · 2017-11-03
**Not working for simple cases**

 I was taking a graduate level machine learning class. In the final course project, we tried to investigate the effects of different regularizations on GAN and WGAN. The two main regularization methods include the one proposed as DRAGAN in this paper and the one proposed in "ON THE REGULARIZATION OF WASSERSTEIN GANS" (referred to as w_reg in the following).

We mainly investigated the following methods: 1a. WGAN with weight clipping, 1b. WGAN with gradient penalty, 1c. WGAN with DRAGAN regularization, 1d. WGAN with w_reg; 2a. GAN, 2b. GAN with DRAGAN regularization.

Since the generated images can only be judged with visualization, besides the image experiments, we also did some experiments on some simple synthetic cases. One exercise is to generate a [-1,1] uniform distribution from a Gaussian distribution. We observed the following results: all the methods 1a-2a are good at generating this simple distribution. However, GAN with the DRAGAN regularization does not. What we observed is that D(x) converges to a function with a hump and therefore all the generated samples are concentrated on a small region, instead of uniform distribution. We adopt a sample code from github. The generator has 2 layers and the discriminator has 1 layer.  The lambda is 10.

We got stuck in the observation for a long time. Later we find out the reason is the regularization term pushes the function to have some slope at the data support, which results in the hump shape. Therefore, the generated samples are mostly concentrated in the region with a large D(x).

I am wondering if the authors have similar experience with the synthetic data experiments? Sometimes, the quality of generated images are hard to judge. Some synthetic data experiments are also needed to verify the performance. Probably I made some mistakes in the code experiments, would be great if the authors can share the code and insights after the review process. My email is leonboellmann0110@gmail.com. Thanks!

---

> ### Author Response · Authors · 2017-11-03
> **Thanks for noting that observation (Edited)**
>
> 1. DRAGAN's regularization is a simple constraint on the discriminator that is used to improve stability. It comes at a cost though! However, as we write at the end of section 2.4, by carefully choosing how you regularize, you can gain stability without losing too much performance.
>
> And though we suggest a hyperparameter setting for mostly image datasets in our paper, it doesn't work for all the distributions possible in all domains. Some tuning is required especially if your domain changes too much (from pixel space) or if the game/players are simple enough, I suggest also reducing the regularization intensity. I think the problem you observed is caused by this (see point 2).
>
> (In hindsight, we should have added a couple of points in Algorithm section to help practitioners use it. We will do so in the final version.)
>
> 2. We show experiments on simple toy datasets in our paper without any issues. Since your domain is [-1,1] (not pixel space) and you are using small networks, I suggest not using default hyperparameters. Reduce 'c' first to something less than 0.1 (say). Further, if you want the best performance, I suggest tuning 'lambda' as well. Understanding what these hyperparameters are doing is essential to use DRAGAN to your advantage - c (size of local regions) and lambda (how much you want to bias). I can take a look at your code or share sample code after the review process :)
>
> Edit: I just realized your training data is uniform in [-1,1]. Your perturbations will be on the manifold in this case, which explains the issue.

---

> > ### Public Comment · ~Leon_Boellmann1 · 2017-11-04
> > **Thanks and suggestions for future research**
> >
> >  I agree that if we downplay the parameter lambda, it will definitely help in this simple case, because it is becoming the original GAN. I think it raises the following questions that need more investigation:
> >
> > 1. Does the GAN structure itself or the regularization term play a more important role in the good performance? This is also one of the objective of our course project and the issue raised in the paper of "Many Paths to Equilibrium: GANs Do Not Need to Decrease a Divergence At Every Step".
> >
> > 2. Does a certain regularization only make sense for particular GAN structures or it can be universally used on top of all existing GAN structures? For this question, it seems the regularization in DRAGAN and the one in the other paper "ON THE REGULARIZATION OF WASSERSTEIN GANS" may only work for WGAN.
> >
> > 3. What is a systematic way of weighing the regularization term and the original objective function, instead of heuristic parameter tuning?
> >
> > 4. What are the pros and cons for all these different kinds of regularizations?

---

> > > ### Author Response · Authors · 2017-11-04
> > > **We do need more research into the GAN game and how regularization helps**
> > >
> > > We make no claims that our regularization term helps find the optimal critic. In fact, we clearly mention that constraints on D actually "hurt" the performance. So, one should use vanilla GANs whenever possible but if you encounter instability, then some form of constraint will help. And we show that DRAGAN can achieve this without losing too much in performance. See end of section 2.4.
> > >
> > > 1. GAN structure is responsible for the good performance, and the constraints are only to improve stability in hard cases.
> > >
> > > 2. We show that DRAGAN helps make the underlying game "easier" in some sense. So, it works with any objective function (see section 3.3), although it might require small amount of hyperparameter tuning.
> > >
> > > 3. There's no easy answer to this. Depends on the game, how strong the players are, domain space and many other factors. But yes, there's need for more research into understanding this and developing better forms of regularization.

---

### Public Comment · ~zihang_zou1 · 2017-12-01
**Should compare with LS-GAN gradient penalty**

I suggest this paper should compare its performance with LS-GAN in experiments.

In Chapter 2.4, this paper refers to LS-GAN and its theorem. Worth to mention that LS-GAN has gradient penalty, which is very flexible since it does not require the norm of gradient to be close to 1 as required in the dual of Wasserstein distance. Instead LS-GAN directly penalizes the gradient as a surrogate of Lipschitz constant derived from its generalization theorem. This makes the gradient penalty different both in theory and in algorithm from that of WGAN-GP. The conclusion in experiments on WGAN-GP thus cannot generalize to LS-GAN. So a direct comparison with LS-GAN is necessary.

You can find the code here, https://github.com/zzzucf/lsgan-gp.

---

> ### Author Response · Authors · 2017-12-09
> **Do take a look at our paper's new revision**
>
> LS-GAN paper has two main ideas:
> 1. Adding a margin
> 2. Making D(x) Lipschitz in data space
> Together, they result in the condition that D(real) - D(fake) ~ ||real - fake||, for any pair.  Our paper credits this idea of imposing gradient constraints (penalties) in the data space to Qi et.al, though its wrongly credited to WGAN-GP paper in the literature.
>
> WGAN-GP provides a different theory/motivation for a very similar constraint (Here, fake can get higher scores than real) or algorithm and does extensive experiments to demonstrate that it helps. So, we only compare with it because essentially they are the same method.
>
> In contrast, DRAGAN only applies constraints in local regions of real samples (we argue for its advantages) and moreover, our paper is focused on developing a novel theory regarding convergence in GANs and mode collapse issue.
>
> And what's suggested in various github pages or later versions of the paper could not be explored or discussed in our paper. We apologize for that

---

### Decision · Program_Chairs · 2018-01-29
**ICLR 2018 Conference Acceptance Decision**

**Decision:**

Reject

**Comment:**

Pros:
The proposed regularization for GAN training is interesting and simple to implement.

Cons:
- Reviewers agree that the methodology is incremental over the WGAN with gradient penalty and the modification is not well motivated.
- Experimental results do not clearly demonstrate the benefits of the proposed algorithm and the paper also lacks comparisons with related works.
GIven the pros/cons, the committee feels the paper is not ready for acceptance in its current state.